Evaluation of unmanned aerial vehicle shape, flight path and camera type for waterfowl surveys: disturbance effects and species recognition

McEvoy John F. johnfmcevoy@gmail.com
Hall Graham P.
McDonald Paul G.
Centre for Behavioural and Physiological Ecology, Zoology, University of New England , Armidale, NSW , Australia
Johnston David
Electronic publication date: 2016 Mar 21
Publication date: 2016
Volume: 4
Electronic Location ID: e1831
Received 2015 Nov 17; Accepted 2016 Mar 1
Copyright: ©2016 McEvoy et al.
Copyright year: 2016
Copyright holder: McEvoy et al.
License: This is an open access article distributed under the terms of the Creative Commons Attribution License, which permits unrestricted use, distribution, reproduction and adaptation in any medium and for any purpose provided that it is properly attributed. For attribution, the original author(s), title, publication source (PeerJ) and either DOI or URL of the article must be cited.
License URL: https://creativecommons.org/licenses/by/4.0/

Keywords: UAV, Aerial survey, Drone, Disturbance, Flight initiation distance

Funding: New South Wales Department of Primary Industry The study was funded by the New South Wales Department of Primary Industry through a grant provided to PM and GH. The funders had no role in study design, data collection and analysis, decision to publish, or preparation of the manuscript.

==============================
The use of unmanned aerial vehicles (UAVs) for ecological research has grown rapidly in recent years, but few studies have assessed the disturbance impacts of these tools on focal subjects, particularly when observing easily disturbed species such as waterfowl. In this study we assessed the level of disturbance that a range of UAV shapes and sizes had on free-living, non-breeding waterfowl surveyed in two sites in eastern Australia between March and May 2015, as well as the capability of airborne digital imaging systems to provide adequate resolution for unambiguous species identification of these taxa. We found little or no obvious disturbance effects on wild, mixed-species flocks of waterfowl when UAVs were flown at least 60m above the water level (fixed wing models) or 40m above individuals (multirotor models). Disturbance in the form of swimming away from the UAV through to leaving the water surface and flying away from the UAV was visible at lower altitudes and when fixed-wing UAVs either approached subjects directly or rapidly changed altitude and/or direction near animals. Using tangential approach flight paths that did not cause disturbance, commercially available onboard optical equipment was able to capture images of sufficient quality to identify waterfowl and even much smaller taxa such as swallows. Our results show that with proper planning of take-off and landing sites, flight paths and careful UAV model selection, UAVs can provide an excellent tool for accurately surveying wild waterfowl populations and provide archival data with fewer logistical issues than traditional methods such as manned aerial surveys.

Introduction

Aerially sourced data is critical to the understanding and census of many ecological systems, such as the use of remotely sensed satellite imagery to investigate the impacts of climate change or the movement ecology of nomadic species (Bartlam-Brooks et al., 2013; Blanco et al., 2008; Felix, 2000; Mueller et al., 2011; Roshier & Rumbachs, 2004) or estimating population sizes using aerial photography (Bako, Tolnai & Takacs, 2014; Trathan, 2004). Research targeting waterfowl populations is no exception. In this field, conservation and management policies concerning agricultural mitigation interventions or harvest seasons are heavily reliant upon accurate population counts that are typically taken from manned aerial surveys, particularly in countries such as Australia where water bodies are often ephemeral and widely dispersed (Kingsford, Porter & Halse, 2011; US Fish and Wildlife Service Division of Migratory Bird Management, 2009). To date, most waterfowl surveys of large water bodies are undertaken using fixed wing aircraft carrying trained observers (Kingsford, 1999; Petrie, Shannon & Wilcox, 2002). The reliability of results from aerial surveys can depend on the experience and training of observers, variation in detectability of different species, and the disturbance caused by flying an aircraft over a wetland at low altitude and high speed (Caughley, 1974; Fleming & Tracey, 2008). While there are many advantages to the use of satellite imagery, ground images can often be obscured by cloud or have poor coverage and be difficult to access for researchers in some countries. Satellite imagery is often sufficient for larger taxa, such as several mammal species on the African savannah (Yang et al., 2014) but is considered unsuitable for smaller taxa such as waterfowl (Conant, Groves & Moser, 2007).

In recent years rapid advances in the quality, availability and range of sensors in commercial unmanned aerial vehicles (UAVs) have lead to their widespread use in the fields of ecology and conservation (Anderson & Gaston, 2013; Chabot & Bird, 2015; Goebel et al., 2015; Jones, Pearlstine & Percival, 2006). UAVs can provide a cheaper, safer and less labour intensive approach compared to traditional aerial surveys. While a single survey flight may cover less ground than a standard aerial survey, they can be used to target specific areas of interest with greater precision and with lower workplace safety risks to employees than travel in manned aircraft entails. Applications include biodiversity assessments (Getzin, Wiegand & Schoning, 2012), counting of colonial species (Grenzdörffer, 2013; Ratcliffe et al., 2015; Sarda-Palomera et al., 2012), observations of target species for conservation management (Brooke et al., 2015; Ditmer et al., 2015; Hodgson, Kelly & Peel, 2013; Koh & Wich, 2012; Vermeulen et al., 2013), and fine scale sensing of habitat composition (Chabot, Carignan & Bird, 2014) and environmental variables in inaccessible terrain (Berni et al., 2009). With the increase in the deployment of UAVs particularly in the study of vulnerable or sensitive species, there is a need to balance the potential disturbance to the animals present with the benefits gained from close observation. The application of UAVs for wildlife surveys is a rapidly advancing field and in 2015 alone there have been several studies that have attempted to quantify the response of animals in wild situations to the presence of an overhead UAV (Chabot, Craik & Bird, 2015; Ditmer et al., 2015; Goebel et al., 2015; Pomeroy, O’Connor & Davies, 2015; Vas et al., 2015). In the studies published to date that have examined this question, researchers have often relied on a single type of UAV (typically a small multirotor model) without comparing potential disturbance effects of the different fixed wing and multi-rotor UAVs that are commercially available. Given the different acoustic profiles, flight patterns and shape of available platforms, it would be unwise to extrapolate a focal species’ response or tolerance between different models of UAVs, such as multirotor versus fixed wing configurations, as animals may react very differently to each type.

A further advantage of UAVs that is seldom raised is that the aerial photography based approach provides a bank of images from which individual species can be independently counted, verified and archived for future analysis or audit. This creates a transparent census technique that increases the usability and cost effectiveness of information gathered if images are made available to other researchers. While aerial photography taken from both manned aircraft and UAVs has proven effective for monitoring large terrestrial mammals (Terletzky & Ramsey, 2014; Vermeulen et al., 2013), identifying much smaller and often more mobile avian taxa to species has thus far proved more challenging. This is particularly true for the identification of waterfowl due to the relative small size of these birds, their camouflaged plumage and similarity in shape and colouration across species (particularly for females and males in eclipse plumage). The acquisition of the high-quality images needed to overcome this challenge has been limited by the resolution and portability of available digital cameras that can be carried by commercially available UAVs.

Recent technological advances appear to have overcome these issues, so the present study had two central aims: (1) To assess disturbance effects on waterfowl from various UAV models that may render a survey invalid, and (2) to assess whether an airborne digital imaging system could provide adequate resolution for unambiguous identification of small bodied waterfowl.

Materials and Methods

This work was carried out under scientific permits from the New South Wales Office of Environment and Heritage (licence no.: SL101457) and was approved by the University of New England Animal Ethics Committee (authority no.: AEC14-104).

Study sites

Field tests of disturbance from UAVs were undertaken at two locations in New South Wales, Australia between March and May 2015 with both sites visited on a total of 6 separate occasions. Little Llangothlin Lagoon (Fig. 1) is a permanent natural wetland in the north east of the state (S-30.086277°, E151.783650°); it is a protected Ramsar wetland that provides an important refuge for numerous waterfowl species during drought. The lagoon is 1 km2 in area with approximately half of the water surface covered with vegetation. The surrounds are characterised by a narrow band of eucalypt dominated open woodland and agricultural pasture (Fig. 1). Whilst the bird community composition did fluctuate between visits, the same species were observed at each survey and included all typical waterfowl for the region. Total bird numbers regularly exceeded 2,000 individuals, with a minimum of 1,000 ducks seen at each visit. The most common waterfowl observed included Eurasian coot (Fulica atra), Pacific black duck (Anas superciliosa), grey teal (Anas gracilis), hardhead (Aythya australis), Australasian shoveler (Anas rhynchotis), pink-eared duck (Malacorhynchus membranaceus), musk duck (Biziura lobata), blue billed duck (Oxyura australis), and black swan (Cygnus atratus). For a complete species list for each site see supplementary material. While birds were present in every area of the lake, distribution of taxa was not uniform; for example dabbling duck species (Anas sp.) were more common near the shore whilst diving birds and swans preferred open water.

Figure 1 Little Llangothlin Lagoon, NSW, Australia.

Approximately 50% of the surface is vegetated and large numbers of birds (>1,000) were distributed across the lagoon. (A) Yellow circles represent take-off sites. White arrows represent approach angles for different flights. (B) One example of a flight path across the lake. Take-off site was away from the edge of the lake and the target altitude was reached before crossing over water. As far as possible any banking or changes in altitude were carried out away from the water. Map data: Google, DigitalGlobe.

Figure 2 Lake Cargelligo, NSW, Australia.

The main lake is approximately 24 km2 and is managed to maintain water levels. The majority of waterfowl in the area congregate on the local sewage works (inset) where all UAV trials were carried out. (A) Yellow circle represents take-off site. White arrows represent approach angles for different flights. (B) One example of a flight path across the sewage works. Take-off site was away from the edge of the water and the target altitude was reached before crossing over water. As far as possible any banking or changes in altitude were carried out away from the water. Map data: Google, DigitalGlobe.

The second site was at the town of Lake Cargelligo in the south east of New South Wales (S-33.313264°, E146.382210°; Fig. 2). This water body is a 24 km2 lake that is managed by local authorities to maintain consistent water levels year round. The main lake supports populations of Australian pelican (Pelecanus conspicillatus) and Australian darter (Anhinga novaehollandiae) but the majority of waterfowl in the region congregate at the adjacent sewage works (Fig. 2). Sewage ponds are rich in nutrients and protected from disturbance and often form a focal point for waterfowl in many areas of Australia (Hamilton & Taylor, 2004; McEvoy et al., 2015; Murray et al., 2012). The sewage works (0.2 km2) consists of constructed ponds of varying turbidity separated by gravel embankments that provided areas for roosting of waterfowl, surrounded by a dense border of Typha spp. rushes. Bird numbers at the sewage works remained stable at approximately 40 individuals at each visit with species including grey teal, pink-eared duck, Pacific black duck, hardhead, and black swan. During the day the majority of waterfowl were found basking on embankments between ponds rather than on the water surface, providing a contrast to the Little Llangothlin Lagoon site. Both study sites were regularly visited by a range of avian predators including white-bellied sea-eagle (Haliaeetus leucogaster), marsh harrier (Circus approximans), and whistling kite (Haliastur sphenurus), all of which prey on small bodied waterfowl (Baker-Gabb, 1984; Marchant & Higgins, 1990). None of the species at either site were observed to be breeding.

Table 1 UAV models used for test flights and their associated characteristics.

UAV	Body shape	Mass (kg)	Take-off	Wingspan/ diameter (m)	Max flight speed (km/hr)	Battery life (min)	
UAVER Avian-P		4.7	Launch Rack + Bungee Cord	1.6	63	60–90	
Skylark II		4	Hand + Bungee Cord	3	40	60–90	
Drone Metrex Topodrone-100		4.5	Launch Rail + Bungee Cord	2	80	60	
DJI Phantom		1.2	Vertical	0.4	10	15	
FoxTech Kraken-130		6	Vertical	1.8	10	15	

UAV models and flights

In total 5 different UAV models of different shapes and sizes were tested in this study to determine if the shape of the model used would elicit different responses from birds (Table 1). These models included fixed wing UAVs with different wing profiles (e.g., delta wing, glider type) and multirotor UAVs from the small, widely used DJI Phantom quadcopter to the larger, more powerful Kraken-130 multirotor UAV (Fig. 3). Each UAV had a unique take-off and landing system, with fixed wing systems requiring a larger clear area for both take-off and landing, launching systems such as bungee cords or rails along with landing airbags, parachutes or nets. The multirotor models could take-off and land vertically from almost any location. Flight time and flight speed also varied between UAV models. Flight time for the fixed wing models was between 60 and 90 min per flight, whereas flight time for multirotor models was often <20 min per flight. Fixed wing UAVs moved at a speed of approximately 40 km/hr while multirotor UAVs moved at approximately 5 km/hr.

Figure 3 Examples of UAV models with different wing profiles.

(A) Avian-P fixed wing UAV, (B) Skylark II fixed wing UAV, (C) Topodrone-100 fixed wing UAV, resembles bird of prey, (D) Kraken-130 multirotor type UAV, (E) Phantom multirotor type UAV, (F) White-bellied sea eagle, a common avian predator active at each study site

At Little Llangothlin Lagoon UAVs were launched from two sites; one within 10 m of the shore and another 500 m from the shore with the latter behind a small rise and thus out of sight of birds on the lagoon. UAVs were launched either directly towards flocks of birds on the lagoon or approached the lagoon at a tangential angle (Fig. 1). Flights were undertaken at a maximum altitude of 120 m and a minimum of 40 m above the lagoon. During flights, each UAV was programmed to fly across the lagoon in a linear north-south direction at a given altitude before changing in altitude by 10 m over land approximately 50 m from the lagoon edge and returning on the opposite north-south route for the next pass (Fig. 1). Altitudes were tested in ascending and descending order to determine if the birds responded differently to the UAV initially entering the area at lower or higher altitudes. Each UAV undertook two test flights for each combination of approach angle and take-off conditions.

At Lake Cargelligo UAVs were launched from a clear site 300 m from the edge of the water out of sight of the birds. Flights were all launched away from the birds due to prevailing wind conditions and flight paths were flown parallel to the embankments where most birds were found. As at Little Llangothlin Lagoon, each UAV carried out a range of passes at ascending and descending altitudes between 120 m and 40 m. Each UAV undertook two test flights for each combination of approach angle and take-off conditions. The Phantom multirotor UAV showed poor results, with birds swimming away from the UAV such that they were not captured on images taken vertically below the unit (Table 2). Further, this unit’s battery life saw it incapable of carrying out a full survey of the lake and as such it was not used in further flights. Due to time constraints, technical problems with the UAV operators’ equipment, and poor weather conditions, some UAVs were not used at both sites (Table 4). The Kraken multirotor UAV was tested at Lake Cagelligo, while the Avian-P, Topodrome-100 and Phantom multirotor were tested at Little Llangothlin Lagoon. The Skylark UAV was tested at both locations. The total number of birds at Lake Cargelligo was smaller (as the water body was not as large) but the same species were recorded and the UAV was tested over a wild, mixed flock of birds.

Table 2 Response of mixed flocks of waterfowl to UAVs of different shapes flying overhead at various altitudes.

For fixed wing UAVs the lower altitudes (15 m) represent take-off where the UAV was launched directly towards the birds before gaining height. NR (green squares) = No discernible response, V (yellow squares) = Vigilance response detected, F (red squares) = Flight response. Cells are marked “N/A” where a given UAV did not fly over birds at that altitude.

UAV	Shape	Altitude Above Water	
		100 m	90 m	80 m	70 m	60 m	50 m	15 m (take-off)	
UAVER Avian-P		NR	NR	NR	NR	V	N/A	F	
Skylark II		NR	NR	NR	NR	V	NA	F	
Drone Metrex Topodrone-100		NR	NR	F	F	F	N/A	F	
DJI Phantom		N/A	N/A	N/A	N/A	N/A	V	V	
FoxTech Kraken-130		NR	NR	NR	NR	NR	V	N/A	

At both sites, disturbance of the birds was monitored during the same flights used to capture digital imagery by two observers at vantage points that covered the entire water body using binoculars, telescopes, and video cameras to observe and record any responses to the UAVs or to natural predators in the area. Disturbance was categorised into three categories: “NR” was used if birds showed no discernible response to the UAV, “V” was used if birds ceased foraging, either orientating or looking towards the UAV or, in some cases, slowly swimming away from the stimulus, or “F” if birds took flight in response to the approaching UAV. No differences in disturbance measures were observed according to whether the UAV was descending or ascending at each pass; hence for simplicity presented results combine both vectors (Table 2).

Digital imaging

Four types of digital camera were mounted to the various UAV models to assess their capability to capture high-quality images of waterfowl against natural backgrounds (Table 3). In all cases the camera was mounted in a gimbal (a pivoted support frame) to allow the camera to remain stable regardless of any turbulence during flight. The physical limitations of each UAV, such as the power to lift camera payloads and attachments, determined the size and shape of camera that could be fitted. Images were digitally recorded to on-board memory cards and downloaded to a laptop in the field upon completion of the flight for preliminary assessment. Adjustments to optimise images via camera setting changes were then made if necessary for subsequent flights. Images from each flight were examined independently by the three authors (all experienced in waterfowl identification) to determine the species present in each image. If there was not unanimous agreement between the three authors as to the species shown in a given image, it was considered to be unusable.

Table 3 Specifications of camera models used in test flights.

Camera	UAV	Focal length	Resolution (megapixel)	Sensor	
Sony RX-1	UAVER Avian-P & Drone Metrex Topodrone-100—Fixed Wing	35 mm	24.3	Full Frame CMOS	
mvBlueCOUGAR-X	Skylark II—Fixed Wing	100 mm	10.1	Full Frame CCD	
Sony A7-R	FoxTech Kraken-130—Multirotor	35–70 mm	36.4	Full Frame CMOS	
Phase 1	FoxTech Kraken-130—Multirotor	80 mm	50	Medium Format CMOS	
GoPro Hero Video Camera	DJI Phantom—Multirotor	21 mm	5.0	CMOS	

Table 4 Disturbance effects on mixed flocks of waterfowl for UAVs launched from different take-off sites and flown at various approach angles.

UAVs were flown either directly perpendicular to a sitting flock of birds or in a tangential flight path running parallel to the main flock of birds. Survey location involved birds at either a large (Llangothlin) or small (Cargelligo) water body.

Location	UAV	No. of flights	Take-off location/direction	Angle of approach	Disturbance effects	
Little Llangothlin Lagoon	Avian-P	2	close to shore/away from lake	Perpendicular to flock	Birds flew away from shore on take-off, no disturbance during flight	
Little Llangothlin Lagoon	Avian-P	2	close to shore/away from lake	Parallel to flock	Birds swam away from shore on take-off. Birds flew during rapid descent to from 80 m to 60 m	
Little Llangothlin Lagoon	Topodrone-100	2	700 m away from shore, out of sight/Towards lake	Parallel to flock	No Disturbance at altitudes above 60 m	
Little Llangothlin Lagoon	Topodrone-100	2	700 m away from shore, out of sight/Towards lake	Perpendicular to flock	Birds flew away from shore on approach at 80 m but became acclimatised to the UAV. Birds flew when the UAV banked and dropped to 60 m.	
Little Llangothlin Lagoon	Phantom	2	close to shore/vertical take-off	Perpendicular to flock	Birds were vigilant and swam slowly away from the UAV	
Little Llangothlin Lagoon	Skylark II	2	close to shore/ directly at flock of birds	Parallel to flock	Birds flew away on take-off, and with banking at 60 m	
Little Llangothlin Lagoon	Skylark II	2	700 m away from shore, out of sight/Towards lake	Perpendicular to flock	No disturbance	
Lake Cargelligo Sewage Works	Skylark II	2	100 m away from shore out of line of sight/away from water	Parallel to flock	No disturbance at any altitude	
Lake Cargelligo Sewage Works	Kraken-130	2	100 m away from shore out of sight/vertical take-off	Parallel to flock	Birds were vigilant and looked up at the UAV at altitudes below 60 m but did not move from their roost.	

Results

Disturbance

Across both study sites and all UAV models, the level of disturbance caused was generally minimal (Table 2). The most extreme category of a flight response by focal waterfowl away from the UAVs was rarely observed but was typically encountered when the UAV was launched directly at a flock of birds at a low altitude during take-off (10–15 m). During the course of the study, resident raptors were observed to fly across both study sites, actively hunting at similar altitudes to those flown by the UAVs. These raptors elicited immediate flight responses from the flocks of waterfowl present with most birds on the lake taking flight as the raptors flew overhead. The flight response to the UAVs, when it did occur, included birds flying away from shore at low altitude for short periods to settle on open water. In contrast, the response to the arrival of a raptor was far more marked with large flocks of birds taking flight and flying for longer periods high above the water before resettling.

The shape and wing profile of each UAV model appeared to influence the response of waterfowl. The UAV with a delta-wing design (Topodrone-100) caused the greatest level of disturbance and flee behaviour (Tables 2 and 4), particularly when it directly approached birds during take-off, or in a direct rather than tangential path (≤80 m altitude), or made a banking manoeuvre while changing altitude (dropping from 70 m to 60 m, for example). These flight periods resemble those of a swooping raptor that is banking to swoop upon prey and the design of the UAV to human eyes was very similar in dihedral angle and shape to that of the larger raptors hunting waterfowl (Fig. 3F).

Disturbance from multi-rotor UAVs was more subtle; the DJI Phantom multirotor when flown at 15 m altitude, resulted in birds swimming away from the UAV when approached. Birds remained vigilant and continuously swam ahead of the field of view of the on-board video camera without taking flight. The larger 8-rotor system (Kraken-130) caused little disturbance at any altitude. The only recorded response occurred at 40 m altitude where roosting birds could be seen to tilt their heads to look up at the UAV, but no further response was noted and birds continued roosting or preening activities (see Pacific black duck in Fig. 4).

Figure 4 Examples of images taken using the Phase-1 medium format digital camera.

Species that are similar in size and shape can be clearly differentiated in photos taken from 60 m above the flock with an 80 mm lens (A and B). Smaller birds such as grebes and black winged stilts can also be easily identified (C and D). A Pacific black duck can be seen tilting its head to look directly up at the camera (C). This image was captured with an ISO = 400, shutter speed = 1/800 s and f-stop = 11. The area footprint of this image is 40 m × 30 m with ground coverage of 5.5 mm/pixel.

Digital imaging

Of the four different camera systems that were trialled only two configurations provided adequate results. The Phase 1 medium format camera equipped with an 80 mm lens and 50 megapixel sensor produced images of high resolution (Fig. 4, ground cover = 5.5 mm/pixel at 60 m altitude) that allowed for the unambiguous identification of very similar species of ducks, as well as smaller non-target species including passerines. The Sony A7-R camera with a 36 megapixel sensor and 50–70 mm lens provided images of comparable resolution to the Phase 1 with a slight but noticeable reduction in resolution (Fig. 5, ground cover = 7.2 mm/pixel at 50m altitude). The other camera systems suffered from either a lack of resolution (e.g., the Sony RX-1, ground cover = 25 mm/pixel at 60 m altitude) and/or technical issues such as motion blur or focusing problems due to software errors or changing light conditions. Unexpected rainfall within a single flight provided images where identification to species level was not possible. For example, the mvBlueCOUGAR-X camera could not be programmed to adjust settings ‘on the fly’ meaning that once programmed prior to takeoff, the camera could not adjust for any change in conditions. This was impractical as changes in light were frequent when a cloud passed in front of the sun, or glare from the water surface reached the lens. As a result, this camera provided very poor-quality images that were not suitable for species identification.

Figure 5 Examples of images taken using the Sony A7-R digital camera.

Species that are similar in size and shape can be clearly differentiated in photos taken from 50 m above the flock with a 70 mm lens (A, B and C). Smaller birds such as swallows and coots can also be easily identified (D and E). The area footprint of this image is 30 m × 20 m with ground coverage of 7.2 mm/pixel. This image was captured with an ISO = 200, shutter speed = 1/640 s and f-stop = 6.3.

Discussion

Many studies have quantified anthropogenic disturbance to wild birds from stimuli such as pedestrians and vehicles (McLeod et al., 2013; Moller et al., 2014). With the increasing popularity of UAVs for ecological research in the past year, there has been a sharp increase in papers assessing the potential for disturbance of wild animals by UAVs (Ditmer et al., 2015; Dulava, Bean & Richmond, 2015; Pomeroy, O’Connor & Davies, 2015; Vas et al., 2015; Weissensteiner, Poelstra & Wolf, 2015). Waterfowl are known to be sensitive to disturbance with relatively high values of flight initiation distance (FID) compared to other species (Bregnballe et al., 2009; Korschgen & Dahlgren, 1992; Madsen, 1995; Weston et al., 2012). Few studies have assessed the disturbance effects of UAVs on waterfowl in wild or natural settings. Studies such as Chabot & Bird (2012) and Vas et al. (2015) used off-the-shelf UAVs to approach flocks of Canada Geese (Branta canadensis) and Snow Geese (Chen caerulescens) (Chabot & Bird, 2012), and semi-wild mallard (Anas platyrhynchos) (Vas et al., 2015) with minimal disturbance recorded. In another recent study by Drever et al. (2015), using a single rotor UAV flying at >60 m altitude, disturbance also appeared to be minimal. With the rapid increase in the application of UAVs for ecological research it is important to gain a broader understanding of their disturbance impacts on different species in wild situations. Our study builds on these results and is the first to directly compare the disturbance effects at two different sized water bodies from a range of camera equipment, including low- to high-end consumer cameras as well as a professional medium-format aerial camera. The UAVs tested in this study cover a range of body shapes and wing profiles that may be deployed for ecological fieldwork. We tested these UAVs in a natural setting with large mixed flocks of different species of water birds and observed that multirotor UAVs had minimal disturbance effects. However, fixed-wing UAVs performed better in collecting aerial photography and were more practical to deploy for larger scale surveys as long as they were flown in a manner that minimised the potential for disturbance.

On repeated flights at varying altitude with different UAV models our results show that if flown with care and attention to potential sources of disturbance, UAVs can prove an effective solution for aerial surveys of waterfowl populations (Table 2). Our findings demonstrate that if take-off and landing occur out of sight of the target species and the UAV has reached its survey altitude before crossing into view of the birds, disturbance should be minimal with in-flight noise quieter than manned fixed-wing aircraft (Fleming & Tracey, 2008). Mixed-species flocks of wild birds were observed to take flight in response to a UAV that was launched from the take-off site toward the flock at low altitude (<40 m). Birds were observed to tolerate a UAV descending from 120 m to 40 m altitude but only if any banking manoeuvres while turning and descending did not occur directly above the flock (Table 4).

It has long been thought that birds, and waterfowl in particular, react differently to silhouettes of predators (e.g., raptors) and non-predators (e.g., geese) flying above them. Early studies (Tinbergen, 1939) found that naïve birds reacted differently to ‘hawk’ and ‘goose’ silhouettes but these findings were later re-assessed (Schleidt, Shalter & Moura-Neto, 2011) and found to be a reaction to novel shapes that disappeared with experience. The outline of the 4 and 8 rotor multirotor UAVs used in this study represented novel shapes that did not resemble any identifiable bird group to our eyes. These UAVs caused no flight response in waterfowl with only a mild swimming response at very low altitudes (15 m). Two of the fixed wing UAVs used in this study, in particular the “Avian-P” model (Fig. 3A), presented an outline that closely resembled that of a non-predatory swan to human observers experienced in waterfowl identification. These ‘glider-type’ UAVs caused no disturbance to large flocks of waterfowl when flown overhead at a steady altitude including flights at the lower limit of the UAV (60 m above water level). The delta-wing type UAV (Fig. 3C) presented an outline resembling raptors that regularly hunt waterfowl at the study sites, particularly when banking and changing altitude, where it caused birds to fly away from the shore toward open water. This is a typical response to a swooping raptor for these species (J McEvoy, pers. obs., 2015). This study afforded us the opportunity to observe responses to actual avian predators as well, including white-bellied sea eagles (Fig. 3F) that have a very similar wingspan and wing shape to the UAVs used. Even though some UAVs had a passing resemblance in silhouette to predators, the arrival of an actual predator resulted in a mass take-off of the mixed flock of birds that was more marked than UAV-evoked responses. Our findings suggest that even though UAVs represent novel objects in the air, wild birds do not react to them as strongly as they do to typical aerial predators.

Legislation governing the use of UAVs in public areas has lagged behind their increasing popularity although some positive changes in regulations have taken place recently (Allan et al., 2015). While small off-the-shelf UAVs with limited range and payload capacity are appropriate for many ecological applications, such as checking the status of nests in hard to reach places (Junda, Greene & Bird, 2015; Potapov et al., 2013; Weissensteiner, Poelstra & Wolf, 2015), larger scale projects require a more specialised and correspondingly larger UAV to carry appropriate equipment and achieve viable flight times (Chabot, Carignan & Bird, 2014). For most researchers this will mean collaborating with a commercial UAV company in order to ensure the technical expertise needed to pilot more complex systems and all relevant aviation permits are in place. While there has been rapid growth in the commercial UAV industry we found that many of the UAV companies are, at least in south-eastern Australia, focused on industrial applications such as mining or civil engineering projects that have vastly different technical requirements to most ecological fieldwork. As a consequence, one of the initial barriers to effectively carrying out aerial surveys of this kind was in communicating effectively with commercial UAV operators the different requirements of working with mobile taxa as opposed to collecting imagery of static geographical features. Future researchers should bear this in mind and have detailed conversations with companies concerning disturbance, approach angles, and take-off and landing sites to avoid costly delays and inaccurate data collection.

High resolution digital images allowing the easy identification of waterfowl species were produced by both the Phase-1 camera (Fig. 4) and the Sony A7-R (Fig. 5) in our trials. Although the images produced by the Phase-1 were of a noticeably higher resolution, the much greater cost (approximately AU$40,000) and size may prove prohibitive for many researchers. Large multirotor UAVs are capable of carrying heavy payloads and can take-off and land vertically in almost any terrain without causing disturbance to waterfowl. However, their current short battery life seriously limits their feasibility for surveys of large areas including even moderately sized water bodies. Single rotor UAVs capable of vertical take-off and landing have been deployed with flight times close to 30 min and faster air speeds than many multirotor systems (Drever et al., 2015) but these UAVs are not as widely available to researchers. Despite this, the low disturbance effects of these UAV designs mean that researchers should consider their use if the current constraints of these models are not too restrictive and a suitable flight time/camera payload can be used.

One of the biggest advantages of using UAVs to collect high-quality digital images is that it allows researchers to archive the results of each survey for future reference, re-assessment, validation, or meta-analysis. While automated systems for counting birds from aerial photographs have been in use for some time (Bajzak & Piatt, 1990), they are generally limited to counting individuals of a single species or colonial birds showing strong contrast with their visual background (Abd-Elrahman, Pearlstine & Percival, 2005; Descamps et al., 2011; Groom et al., 2011; Trathan, 2004). The task of automatically identifying species of similar size, shape, and colouration from large mixed flocks remains a serious challenge to overcome. Developing versatile algorithms to identify and count a variety of waterfowl species in different habitat settings will be a key area of future research in this field.

When attempting to accurately identify waterfowl to species level using UAV photography there are many trade-offs that must be considered. Ideally, using a very high resolution camera such as the Phase 1 medium format camera (50 megapixels) allows the UAV to fly at higher altitudes covering a large area footprint (40 m × 30 m at 60 m altitude) with a medium focal length lens (80 mm) and still gather images of suitable quality (Fig. 4). Due to constraints of budget or the maximum payload of the available UAVs, it may not be possible to fly with a large, high resolution camera such as the Phase 1 camera used in this study. Problems with lower resolution could be overcome by flying the UAV lower or by using longer lenses, but this needs to be balanced against potential disturbance as well as a reduction in the overall footprint covered by each image. We also observed an increased potential for image blurring when using longer lenses at lower altitudes. We used the Sony A7-R camera (36 megapixels) with a 70 mm lens at 50 m altitude and captured images that allowed for easy identification of waterfowl to species (Fig. 5) with a slight reduction in the area footprint of each image (30 m × 20 m). Further trade-offs occur with the choice of UAV when considering the ability of the UAV to survey a large water body in a short period of time, the ability to carry the necessary camera equipment, and whether the shape, wing profile, or flight behaviour of the UAV are likely to disturb the target animals.

Based on our findings, the best results for successfully surveying a medium sized (>1 km2) wetland and easily identifying waterfowl would come from a combination of a digital camera with a minimum 36 megapixel full-frame sensor combined with a fixed wing UAV with long straight wings to cover large wetlands in a single survey. That is not to say that other combinations of UAVs and cameras are unsuitable, but that this combination appears to offer the longest flying time combined with high-quality imagery suitable for species identification at an approximate resolution of 7.2 mm per pixel that was required to achieve our desired accuracy in identification. Prior to deployment, careful consideration should be given to all of the trade-offs mentioned above before embarking on a waterfowl survey using UAVs as different conditions or availability of hardware may necessitate a slightly different approach. Where possible, the take-off and landing sites should be carefully selected to remain out of sight of the target birds and flight paths should be programmed to approach tangentially to the main flock of birds. Where possible, any turning manoeuvres or sharp drops in altitude should be performed away from the shore with the UAV flying at a fixed altitude across the study area to minimise any potential for disturbance.

Supplemental Information

Supplemental Information 1 List of bird species at each study sites

List of bird species observed at two study sites, Little Llangothlin Lagoon and Lake Cargelligo sewage works, NSW between March and May 2015.

Click here for additional data file.

UAV surveys were carried out with the assistance of a number of commercial companies: UAV Geomatics Australia, Airsight Australia, Flight Data Systems, and UAViation Australia. We are indebted to the National Parks and Wildlife Service, NSW and Lachlan Shire Council for their assistance with accessing study sites. We are grateful to a number of people for their assistance with field work: Koen Dijkstra, Milla Mihailova, Ahmad Barati, Louise Streeting, Sigrid Mackenzie and Alice Bauer.

Additional Information and Declarations

Competing Interests

Author Contributions

Animal Ethics

Field Study Permissions

Data Availability

The authors declare there are no competing interests.

John F. McEvoy conceived and designed the experiments, performed the experiments, analyzed the data, wrote the paper, prepared figures and/or tables, reviewed drafts of the paper.

Graham P. Hall conceived and designed the experiments, performed the experiments, reviewed drafts of the paper.

Paul G. McDonald conceived and designed the experiments, performed the experiments, contributed reagents/materials/analysis tools, reviewed drafts of the paper.

The following information was supplied relating to ethical approvals (i.e., approving body and any reference numbers):

This work was approved by the University of New England Animal Ethics Committee (authority no: AEC14-104).

The following information was supplied relating to field study approvals (i.e., approving body and any reference numbers):

This work was carried out under scientific permits from the New South Wales Office of Environment and Heritage (licence no.: SL101457).

The following information was supplied regarding data availability:

The research in this article did not generate any raw data beyond what is presented in the manuscript text.

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
