# Peer review of "Evaluation of unmanned aerial vehicle shape, flight path and camera type for waterfowl surveys: disturbance effects and species recognition"

_PeerJ, doi:10.7717/peerj.1831_

## Round 0.1 · original submission · Major Revisions

Three reviews of your paper have been received, and all were favorable. The common thread amongst the reviews was that the manuscript in question would be improved by a more recent assessment of the literature on UAVs in wildlife studies, and the inclusion of any relevant findings they may include. I encourage you to pay close attention to these reviews and in particular the review provided by Reviewer #2. The comments and critiques provided by this review will greatly strengthen your manuscript and make it more current - something that we can aspire to with online journals.

·

Basic reporting

A few citations of new publications should be added to the introduction, this is a fast moving field of study, these new citations will help your publication.

Dominique Chabot, David M. Bird. Wildlife research and management methods in the 21st century: Where do unmanned aircraft fit in?Journal of Unmanned Vehicle Systems, 2015, 3:137-155, 10.1139/juvs-2015-002

James Junda, Erick Greene, David M. Bird. Proper flight technique for using a small rotary-winged drone aircraft to safely, quickly, and accurately survey raptor nests. Journal of Unmanned Vehicle Systems, 2015, 3:222-236, 10.1139/juvs-2015-0003

Experimental design

I would like to see which cameras were used on which aircraft.

You should consider adding the track of the entire flight path to Figures 1 and 2. This would help the reader visualize how each flight was accomplished.

Validity of the findings

No Comments

Additional comments

A well done and useful addition the the growing field of UAV driven wildlife research.

·

Basic reporting

I have pre-written a traditional "line-by-line" review of the manuscript which unfortunately doesn't readily break down among these boxes, though comments relating to basic reporting, experimental design, and validity of the findings are contained throughout. I therefore hope it is acceptable to simply paste my entire review into the "General Comments for the Author" box.

Experimental design

Some additional details, clarifications and revisions are required in the Materials/Methods and Results sections and associated tables; please see "General Comments for the Author".

Validity of the findings

Overall the findings are of value and interest, however there is a tendency to overstate their robustness and absoluteness throughout the manuscript; please see "General Comments for the Author".

Additional comments

Thank you for the opportunity to review this manuscript. I believe it presents timely research that will be of interest to the scientific community, as the use of UAVs to survey waterbirds and other wildlife appears to be on the brink of exploding. My main criticisms of the manuscript, as detailed in my specific comments below, are that: (1) there is some lack of detail and clarity in the methods and results; and (2) the breadth and rigour of the conducted research as well as the “absoluteness” of the results/conclusions seem to be somewhat overstated at various points throughout the text, and a more humble/qualified account of what was accomplished and observed would benefit the manuscript without diminishing the value and interest of the findings. I hope my specific comments below will be helpful towards improving the manuscript.

General comment: Although I believe your manuscript contains useful and novel information that will be of interest to the scientific community, it’s worth noting that there’s been a sudden explosion of papers on UAV surveys of waterbirds in the past year or so. Aside from those you’ve already cited, I invite you to also take a look at the following papers if you haven’t already done so, and consider citing/discussing them as well in the interest of making your manuscript as current as possible. Some of them also assessed subject reactions to the UAVs to some extent, as well as the image resolution required to ID species. I’m not saying you need to cite all of these papers; just sharing what is to my knowledge the comprehensive list of primary literature on surveying waterbirds with UAVs in case there are some relevant ones that you may have overlooked:

Drever, M.C., Chabot, D., O’Hara, P.D., Thomas, J.D., Breault, A., and Millikin, R.L. 2015. Evaluation of an unmanned rotorcraft to monitor wintering waterbirds and coastal habitats in British Columbia, Canada. J. Unmanned Veh. Syst. 3(4): 256–267.

Dulava, S., Bean, W.T., and Richmond, O.M.W. 2015. Applications of unmanned aircraft systems (UAS) for waterbird surveys. Environ. Pract. 17(3): 201–210.

Goebel, M.E., Perryman, W.L., Hinke, J.T., Krause, D.J., Hann, N.A., Gardner, S., and LeRoi, D.J. 2015. A small unmanned aerial system for estimating abundance and size of Antarctic predators. Polar Biol. 38(5): 619–630.

Ratcliffe, N., Guihen, D., Robst, J., Crofts, S., Stanworth, A., and Enderlein, P. 2015. A protocol for the aerial survey of penguin colonies using UAVs. J. Unmanned Veh. Syst. 3(3): 95–101.

And here are few more older ones that you may or may not have already come across:

Chabot, D., and Bird, D.M. 2012. Evaluation of an off-the-shelf unmanned aircraft system for surveying flocks of geese. Waterbirds. 35(1): 170–174.

Grenzdorffer, G.J. 2013. UAS-based automatic bird count of a common gull colony. Int. Arch. Photogramm. Remote Sens. Spatial Inf. Sci. XL-1(W2): 169–174.

Sarda-Palomera, F., Bota, G., Vinolo, C., Pallares, O., Sazatornil, V., Brotons, L., Gomariz, S., and Sarda, F. 2012. Fine-scale bird monitoring from light unmanned aircraft systems. Ibis. 154(1): 177–183.

Watts, A.C., Perry, J.H., Smith, S.E., Burgess, M.A., Wilkinson, B.E., Szantoi, Z., Ifju, P.G., and Percival, H.F. 2010. Small unmanned aircraft systems for low-altitude aerial surveys. J. Wildl. Manage. 74(7): 1614–1619.

Line 65: Contrary to what is implied in this sentence, it’s my understanding that Terletzky & Ramsey (2014) collected their aerial imagery from a conventional fixed-wing aircraft, not a UAV.

Table 1: The “Body Shape” column seems redundant with the actual photos of the UAVs in Figure 3. I suggest removing this column and perhaps simply specifying in the table which models were fixed-wing and which were multirotor.

Line 146: Is it normal that reference to Table 4 appears in the text before reference to Table 2?

Lines 150–151: “"NR" saw no discernible response being observed from the focal animals” is awkwardly formulated; please rephrase.

Line 158: I believe the reference here should be to Table 3, not Table 2.

Table 3: I suggest adding a column listing which UAV(s) each of the cameras was mounted on, as this is not properly detailed in the text and quite relevant to report.

Lines 156–168: (1) Consider providing more detailed information on the shooting settings you used, e.g. shooting mode, shutter speeds, aperture and ISO settings, focus settings, etc. I realize this could get a bit tedious with four different cameras having been used, but I think at least some basic/summary info would be helpful for new/prospective UAV users who, in my experience, in the absence of guidelines are susceptible to employing inappropriate shooting settings and consequently ending up with suboptimal imagery. You could perhaps alternatively add these details to Table 3. (2) Furthermore, in the absence of any information to the contrary, are readers to assume that camera imagery was captured over the course of the same flights during which disturbance to birds was assessed?

Lines 171–172: Delete either “general” (line 171) or “generally” (line 172) from this sentence.

Table 4: I’m a bit confused by this table. Is this a comprehensive list of all the individual flights you performed over the course of your study? If so, I’m having trouble reconciling its contents with the statements in the methods section to the effect that “each UAV undertook two test flights for each combination of approach angle and take off conditions”, which you state for each of your two study sites. I do note that you add that “some UAVs were not used at both sites”, but you also mention at the beginning of the Study Sites section that “each site [was] visited on 6 separate occasions”, so overall there definitely seems to be a lack of clarity/accord among these various statements and the contents of Table 4. Some clarification in the text and/or the table is in order.

Lines 191–198: Judging by Table 4 (which I admittedly may be misinterpreting), it appears as though you only flew the Phantom a single time, at Little Llangothlin Lagoon, and the Kraken-130 a single time, at Lake Cargelligo Sewage Works. If this is indeed the case, you need to emphasize that the observations described in this paragraph should be interpreted with caution due to the small number of flights performed with the multirotor UAVs compared to the fixed-wing models. It is particularly noteworthy that you only appear to have flown the Kraken-130 at Cargelligo, presumably over relatively small numbers of birds (as per your description in the Study Sites section), in contrast to the large numbers of birds at Llangothlin where you flew all the other UAV models.

Line 202: I suggest inserting a reference to Figure 4 after “images of high quality”.

Lines 204–205: Saying that the Sony camera provided “images of comparable quality to the Phase 1 with only a slight reduction in picture quality” is not terribly informative. On what criteria did you base your assessment of picture quality? If you simply mean that the resolution was slightly lower, then say so.

Line 206–208: These rather ambiguous comments highlight the need to better detail the shooting settings you used in the methods section. “Motion blur” suggests that these cameras were perhaps incapable of sufficiently fast shutter speeds, while alluding to “inadequate autofocus capabilities” is a bit puzzling since you should generally always shoot aerial photos from a UAV with the focus set to infinity, except perhaps if you’re flying very low (e.g. <100 ft).

Lines 211–213: “However, few studies have assessed the disturbance impacts of UAVs, despite their increasing popularity for ecological research in recent years.” Until quite recently this statement was true, but not so much anymore following a recent sharp increase in the number of published papers on use of UAVs to survey wildlife, many of which assessed disturbance to some degree. Aside from Chabot et al. (2015) and Vas et al. (2015) which you cite in the introduction, among the other UAV-waterbird papers I listed above, Drever et al. (2015), Dulava et al. (2015) and Goebel et al. (2015) also assessed disturbance in some manner. Beyond these, the following recent papers have also assessed UAV disturbance to various wildlife species:

Brooke, S., Graham, D., Jacobs, T., Littnan, C., Manuel, M., and O’Conner, R. 2015. Testing marine conservation applications of unmanned aerial systems (UAS) in a remote marine protected area. J. Unmanned Veh. Syst. 3(4): 237–251.

Ditmer, M.A., Vincent, J.B., Werden, L.K., Tanner, J.C., Laske, T.G., Iaizzo, P.A., Garshelis, D.L., and Fieberg, J.R. 2015. Bears show a physiological but limited behavioral response to unmanned aerial vehicles. Curr. Biol. 25(17): 2278–2283.

Pomeroy, P., O’Connor, L., and Davies, P. 2015. Assessing use of and reaction to unmanned aerial systems in gray and harbor seals during breeding and molt in the UK. J. Unmanned Veh. Syst. 3(3): 102–113.

Weissensteiner, M.H., Poelstra, J.W., and Wolf, J.B.W. 2015. Low-budget ready-to-fly unmanned aerial vehicles: An effective tool for evaluating the nesting status of canopy-breeding bird species. J. Avian Biol. 46(4): 425–430.

I think the disturbance study presented in your manuscript is nevertheless a welcome addition to this growing body of literature, but you might simply change the wording of this sentence to something down the lines of: “As the use UAVs for wildlife surveys increases, it is of interest to gain a broader understanding of their disturbance impacts on animal subjects.” I’m now realizing you might also consider revising some of the text in the Introduction (starting at line 51) in a similar manner.

Line 218: I believe the citation should be “(Vas et al. 2015)”.

Lines 218–220: I’m concerned that the wording of this sentence overstates what you actually accomplished in your study. Again based on Table 4, there appears to be only a single UAV out of the five that you in fact flew at both sites. Furthermore, the wording “across different sized water bodies” embellishes the fact you conducted your study over precisely two water bodies, one large and one small. Finally, the wording “a range of medium sized UAVs used to carry sophisticated camera equipment” seems at odds with the fact that only two of the four cameras you tested were deemed to be adequate and your criticisms of the other two certainly don’t make them out to be terribly sophisticated… Unless of course all five UAVs were able to carry one or the other of the two adequate cameras, although I suspect this is not the case; either way, you need to specify in the methods/table which cameras were mounted on which UAVs, as I mentioned above.

Line 223: Suggest changing to: “and we observed that multirotor UAVs had minimal…”

Line 229: Remove “low cost”, since there is no other mention or discussion in the manuscript of the cost of purchasing and operating the UAVs you used; not to mention the additional time/cost of identifying and counting birds in the recorded imagery after field work has been completed, which is too often overlooked when assessing the cost-effectiveness of UAVs.

Line 232: “much less noisier” :(

Lines 248–249: “… down to the lowest altitude the UAV was capable of flying of 60m above water level” is awkwardly formulated; please rephrase.

Lines 258–259: Suggest changing “represented” to “represent” and changing the end of the sentence to: “… do not react to them as strongly as they do to actual predators.”

Line 260–277: Overall excellent discussion of practical considerations for prospective UAV users in the wildlife/ecology community.

Line 280: Note that earlier you state that images from the Sony camera had a “slight reduction in picture quality” compared to the Phase-1, while here you state that images from the Phase-1 camera were of “noticeably higher quality”.

Lines 282–285: (1) Use the present tense throughout this sentence. (2) Consider mentioning that there are also heavy-lift single-rotor VTOL UAVs available which, although not as popular and widespread as their multirotor counterparts, are actually more energy-efficient and consequently tend to boast longer flight times. For example, such a model was used by Drever et al. (2015) to survey waterbirds, capable of over 30 minutes of flight time at relatively fast cruising speeds of up to ~70 km/h.

Line 285: Change “lack of disturbance” to “low disturbance”.

Line 287: Remove “logistical”.

Line 288–297: Very good of you to bring up the issue of automated detection and counting of subjects in the imagery. This relates to my earlier comment about the need to factor post-field-work analysis of the imagery—which often entails lengthy/tedious manual interpretation—into any statement about the cost-effectiveness of UAVs.

Lines 296–297: Suggest reformulating the sentence as follows: “Developing versatile algorithms to identify and count a variety of waterfowl species in different habitat settings will be a key area…”

Line 299: I find the simple recommendation of a minimum-36-megapixel camera to be somewhat impractical and overly restrictive, since what is ultimately required to successfully ID bird species is a certain ground resolution (i.e. in cm/pixel). By any chance did you determine the ground resolution required to ID birds in your study? If so, this would be more useful to report. Lower-megapixel cameras can achieve equivalent ground resolutions to higher-megapixel cameras if simply flown lower, or by using longer focal lengths (i.e. zooming in more). Of course, flying lower may increase disturbance to birds, and both flying lower and zooming in more will decrease the area footprint of photos. These are points that you could add to your discussion in order to make it more insightful. Overall I agree that it’s desirable to use a very high-resolution full-frame camera, if possible, so that you can achieve a high ground resolution from a relatively high altitude that both minimizes disturbance and maximizes the area covered by each photo; but I think you need to do a better job of discussing these interrelated variables and trade-offs. Otherwise, I’m concerned that naïve readers might get the impression that they won’t be able to successfully survey waterfowl with a UAV unless they can get their hands on a 36-megapixel camera (not to mention a UAV that can lift one!), which is not necessarily the case.

Line 300: (1) What do you consider to be “lightweight”, and are such models actually capable of lifting a high-resolution (e.g. 36 MP) camera? (2) As with the camera recommendation, I find the recommendation of a lightweight fixed-wing UAV with long straight wings in order to achieve a “successful” waterfowl survey to be overly restrictive. At most you could perhaps state that among the UAVs you tested, this is the type that offered the best overall combination of long flight time, low disturbance and suitable-quality imagery; but based on my own experience, I’m confident that waterfowl surveys can be successfully accomplished with other types of UAVs as well.

Lines 301–303: As with the next sentence that begins with “Where possible”, I suggest you add a “where possible” to this sentence as well, as there will indeed not always be a launching/landing site available that is out of sight of the birds. As an aside, I think it’s important to be sensible about disturbance and generally convey that although it’s always desirable to minimize it to the extent possible/reasonable, there will inevitably be cases where some amount of disturbance to wildlife subjects is unavoidable, and that it isn’t the end of the world so long as the value of the data being collected is judged to outweigh any temporary stress imposed on the subjects; in particular if alternative survey methods cause even more disturbance or are significantly less efficient or accurate. You do appropriately allude to this “balance” in the Introduction (lines 49–51).

Thanks again for the opportunity to review this manuscript, and I hope that my feedback will prove helpful.

Best regards,

Dominique Chabot

·

Basic reporting

see general comments

Experimental design

see general comments

Validity of the findings

see general comments

Additional comments

General comments
This paper examines how the use of UAVs affect roosting and feeding waterfowl, and may be the first to compare different models (fixed wing vs. multirotor) on the same set of species. A series of trials was conducted during March-May 2015 to take photographs of several waterfowl species at two sites in New South Wales, Australia, and to record responses to UAVs at different heights from 50m to 100m approaching the birds at different angles. The authors find that waterfowl remained unresponsive when UAVs were flown above >40m for multirotor UAVs and >60m for fixed-winged models, and that with diligent use during take-off and landings, complete photograph surveys could be conducted with minimal disturbance to waterfowl over large wetlands. This research offers novel and valuable insights into the use of an emergent technology for enumerating wildlife populations and will greatly interest the readership of PeerJ.
To improve the manuscript, I suggest the following:
The Abstract needs a few more details about where and when the trails were conducted.
Contrasts between the use of UAVs and traditional aerial surveys for waterfowl should be done with caution since this paper did not involve traditional surveys, and so comparisons of cost and safety should be tempered with this in mind. Traditional aerial surveys may lack the precision available with UAVs, but they offer an opportunity to cover much larger areas, especially if the crew is experienced. In North America, fixed- and rotary-wing aerial surveys for waterfowl have been conducted very successfully for decades to provide information for harvest management (Smith 1995). It may be best to simply present the types of information that can be gathered with UAVs, and let managers decide what costs and risks they wish to incur. A great benefit of UAVs is the ability to simultaneously collect detailed information on waterfowl abundance and habitat condition (e.g., percent emergent vegetation), so perhaps this potential could be elaborated.
The critical detail for identifying birds from photographs is the ground resolution (cm/pixel). Would it possible to calculate for the images in this study? I realize that with 4 cameras and 6 heights, this involves 24 different resolutions, but it would good to provide this measure for comparisons with other studies.
The paper contains many comma errors. Please review the use of commas (e.g., https://owl.english.purdue.edu/owl/resource/607/02/), and correct accordingly.
The following numbers refer to Line number.
77-110. Please provide an indication of how waterfowl use this site. I assume March-May is the austral non-breeding season, but it’d be good to state this explicitly, since we can reasonably expect that ducks will respond differently during other times of the year.
152-164. can you provide the shutter speed, and the frequency of photographs taken, e.g., 1 per sec?
209-212. Please correct the citation. Drever et al. (2015) also examined UAV disturbance in waterfowl.
Table 1 and 2. The use of shape is a great idea.
Figures 1 and 2. These images do not reproduce well in black and white, especially the red lines which are lost in the background. Could they be modified to improve their appearance in black and white?

---

## Round 0.2 · accepted · Accept

I have read the revised manuscript along with the final reviews. Your revisions and additions have greatly strengthened the manuscript. At this point it is important to pay attention to the final editorial comments provided by reviewer #2 as the manuscript is prepared for publication. I'm really pleased to see this paper progress.

·

Basic reporting

All of my concerns were addressed and the manuscript along with this and other much more extensive changes lead me to lend my approval to the publication of this manuscript.

Experimental design

All of my concerns were addressed and the manuscript along with this and other much more extensive changes lead me to lend my approval to the publication of this manuscript.

Validity of the findings

All of my concerns were addressed and the manuscript along with this and other much more extensive changes lead me to lend my approval to the publication of this manuscript.

Additional comments

These changes have greatly benefited this manuscript. It is well done research, presented well and adds relevant knowledge. Good job.

·

Basic reporting

See general comments for the author.

Experimental design

See general comments for the author.

Validity of the findings

See general comments for the author.

Additional comments

I congratulate the authors for doing an overall thorough and satisfactory job of addressing my comments as well as those of the other reviewers; the manuscript is now significantly improved. I suggest only a few further minor revisions:

- The matter of which UAVs were flown at each site is now generally much clearer. However, I note that on line 161 (Word file) you state that the Phantom was flown at Lake Carpelligo, whereas Table 4 indicates that it was flown at Little Llangothlin.

- Line 254 (Word file): Change “across two different sized water bodies” to “at two different sized water bodies”.

- Line 256 (Word file): The Phase One is the only true aerial photogrammetric camera you used, which I would label as “professional” rather than “commercial”, while all of the others are consumer-grade models, albeit ranging in performance/sophistication. I therefore suggest rephrasing as follows: “… a range of medium-sized UAVs carrying a range of camera equipment, including low- to high-end consumer cameras as well as a professional medium-format aerial camera.”

- Line 294 (Word file): Add a comma after “air” and delete “above them”.

- Line 333 (Word file): Note that Rodgers et al. (2005) did not in fact use any automated bird counting methods or even aerial photography—only human observers onboard the aircraft—so I suggest deleting this citation.

- As originally pointed out by Reviewer 3, I remain concerned about the occurrence of comma errors in the revised manuscript. Hopefully the journal’s editorial staff can assist in making the necessary corrections during the final production stage.

Thanks again for the opportunity to review this manuscript, which I believe will make a valuable contribution to the literature.

Best regards,

Dominique Chabot